# U-Net Based Segmentation and Characterization of Gliomas

**DOI:** 10.3390/cancers14184457

**Published:** 2022-09-14

**Authors:** Shingo Kihira, Xueyan Mei, Keon Mahmoudi, Zelong Liu, Siddhant Dogra, Puneet Belani, Nadejda Tsankova, Adilia Hormigo, Zahi A. Fayad, Amish Doshi, Kambiz Nael

**Affiliations:** 1Department of Radiology, Icahn School of Medicine at Mount Sinai, New York, NY 10029, USA; 2Department of Radiological Sciences, David Geffen School of Medicine at University of California Los Angeles, Los Angeles, CA 90033, USA; 3Biomedical Engineering and Imaging Institute, Icahn School of Medicine at Mount Sinai, New York, NY 10029, USA; 4Department of Pathology, Icahn School of Medicine at Mount Sinai, New York, NY 10029, USA

**Keywords:** deep learning, glioma, tumor segmentation, radiogenomic, isocitrate dehydrogenase 1, magnetic resonance imaging

## Abstract

**Simple Summary:**

Gliomas comprise 80% of all malignant brain tumors. We aimed to develop a deep learning-based framework for the automatic segmentation and characterization of gliomas. In this retrospective study, patients were included if they: (1) had a diagnosis of glioma confirmed by histopathology and (2) had preoperative MRI with the inclusion of FLAIR imaging. The deep learning-based U-Net framework was developed based on manual segmentation on FLAIR as the ground truth mask for automatic segmentation and feature extraction, which were used for the prediction of biomarker status and prognosis. A total of 208 patients were included from our internal dataset with stratified sampling to split the database into training and validation. An external dataset (*n* = 31) from an outside institution was used for testing. The dice similarity coefficient of the generated mask was 0.93 on the testing dataset. The prediction of the radiomic model achieved an AUC of 0.88 for IDH-1 and 0.62 for MGMT on the testing dataset. Our deep learning-based framework can detect and segment gliomas with excellent performance for the prediction of IDH-1 biomarker status and survival.

**Abstract:**

(1) Background: Gliomas are the most common primary brain neoplasms accounting for roughly 40–50% of all malignant primary central nervous system tumors. We aim to develop a deep learning-based framework for automated segmentation and prediction of biomarkers and prognosis in patients with gliomas. (2) Methods: In this retrospective two center study, patients were included if they (1) had a diagnosis of glioma with known surgical histopathology and (2) had preoperative MRI with FLAIR sequence. The entire tumor volume including FLAIR hyperintense infiltrative component and necrotic and cystic components was segmented. Deep learning-based U-Net framework was developed based on symmetric architecture from the 512 × 512 segmented maps from FLAIR as the ground truth mask. (3) Results: The final cohort consisted of 208 patients with mean ± standard deviation of age (years) of 56 ± 15 with M/F of 130/78. DSC of the generated mask was 0.93. Prediction for IDH-1 and MGMT status had a performance of AUC 0.88 and 0.62, respectively. Survival prediction of <18 months demonstrated AUC of 0.75. (4) Conclusions: Our deep learning-based framework can detect and segment gliomas with excellent performance for the prediction of IDH-1 biomarker status and survival.

## 1. Introduction

Radiogenomic mapping has emerged as a promising noninvasive tool for the successful prediction of glioma biomarkers, which has important implications for understanding pathophysiology and creating targeted treatment regimens [1]. Some of the most studied biomarkers include isocitrate-dehydrogenase-1 (IDH1) [2], methylguanine-DNA-methyl-transferase (MGMT) promoter methylation profile [3], transcriptional regulator (ATRX) [4], and epidermal growth factor (EGFR) [5]. In particular, IDH-1 and MGMT have emerged as important biomarkers, with IDH-1 shown to be an independent positive prognostic biomarker correlating with longer progression-free survival and positive treatment outcome for chemoradiotherapy [6]. MGMT gene promoter methylation has also been shown to predict treatment outcomes for temozolomide plus radiotherapy [7,8].

Most recently, conventional MRI sequences such as T1-weighted postcontrast (T1c+), fluid-attenuated inversion recovery (FLAIR), and diffusion imaging [9] have been found to be useful in predicting IDH-1 mutation [10,11,12,13], MGMT methylation status [7,8], EGFR overexpression [14,15], ATRX mutation [12,16], PTEN deletion [17], and TP53 mutation [13]. The ability to predict biomarker statuses noninvasively is invaluable as large tissue specimens requiring multiple biopsy attempts are often needed for confirmatory histopathological diagnoses. Radiomic association with survival prediction has also been underway. Carlson et al. previously demonstrated that in patients with malignant gliomas, vascular endothelial growth factor (VEGF) status was predictive of patient survival independent of edema [18]. Eliat et al. incorporated texture analysis with dynamic contrast-enhanced MRI to differentiate malignant glioneuronal tumors, which are associated with increased survival and less metastasis, from glioblastoma multiforme (GBM) [19]. More recently, Rao et al. used MRI sequences to demonstrate that volume-class, hemorrhage, and T1/FLAIR-envelope ratio could stratify survival in patients with GBM [20].

In recent years, models developed as part of the Brain Tumor Segmentation (BraTS) challenges have been shown to successfully detect and segment gliomas [21,22]. Several deep learning methods, such as single multi-task network (OM-Net) [23], 3D convolutional neural networks applying hierarchical segmentation [24], and 3D U-Net architecture [25] have shown promising results in glioma segmentation. Extensive research is underway to predict biomarkers from these segmentations; however, there is scarce data on combining segmentation with radiomic extraction for the prediction of biomarkers and survival prediction.

In this study, we aimed to develop a U-Net-based, fully automated framework for segmentation and characterization of gliomas using MR radiomics for predicting biomarker status and prognosis. We further assess the applicability of our algorithm using external validation.

## 2. Materials and Methods

### 2.1. Patient Selection

This retrospective study was approved by an institutional review board and informed consent was waived. Patients with initial diagnoses of glioma between January 2016 and September 2020 were reviewed. Patients were included if they (1) had a diagnosis of gliomas with known IDH-1 and MGMT statuses from surgical histopathology; (2) had preoperative MRI including FLAIR within 30 days of biopsy or surgical resection. Furthermore, any patients with prior chemotherapy or resection were excluded. MR with motion artifacts were excluded. The patient’s survival defined from the date of diagnosis (surgical pathology) onward was documented when available. The survival was dichotomized to poor versus good using an 18-month cutoff as the high-end of the survival curve in the era of modern glioma treatments [26]. The internal dataset was collected from Mount Sinai Hospital (MSH) for training and validation purposes. Developed models were tested on an external dataset from the University of California, Los Angeles (UCLA).

### 2.2. Histopathological Data

Tumor tissue samples were obtained from biopsy or resection as part of routine diagnostic neuropathology and molecular evaluation. IDH-1 (specifically IDH1-R132H) was assessed using immunohistochemistry. MGMT promoter methylation was assessed using pyrosequencing of bisulfite-treated genomic DNA (CpG sites 74–79, QIAGEN).

### 2.3. Image Acquisition

Image acquisition was performed using a standardized preoperative brain tumor MRI protocol within our radiology department. The imaging protocol for FLAIR images were repetition time (TR)/echo time (TE)/inversion time (TI), 8000–12,000/98–130/2400–2700 ms.

### 2.4. U-Net Based Auto-Detection and Segmentation of Gliomas

A DenseNet121 [27] based U-Net framework pretrained on the RadImageNet [28] dataset was developed based on symmetric architecture from the 512 × 512 segmented maps from FLAIR as the ground truth mask. The input size was 256 × 256 × 3, where all grayscale images were converted to RGB images in order to use pretrained weights with all three channels set to the same value. The ReLU activation function was implemented on each convolutional layer. A batch size of 16, RMSProp with a learning rate of 0.001, and binary cross entropy loss function were used. Stratified sampling was performed to split the database into training (*n* = 176), validation (*n* = 32), and testing (*n* = 31). Dice similarity coefficient (DSC) was calculated to assess the overlap of the deep learning-based segmentation map and ground truth segmentation divided by the total size of the two masks.

### 2.5. Volume Acquisition and Texture Analysis

Tumor segmentation was performed manually using volume-of-interest (VOI) analysis on commercially available FDA-approved software (Olea Sphere software, Olea Medical SAS, La Ciotat, France). The entire tumor volume, including FLAIR hyperintense infiltrative component and necrotic and cystic components, was manually segmented by an expert neuroradiologist (K.N., 10 years) on FLAIR images (Figure 1). A total of 95 texture features were extracted from predictive masks generated by the aforementioned U-Net via the pyradiomics [29] package in Python 3.8.10. These included 2 shape features, voxel volume and surface volume ratio, 18 first-order metrics, such as the mean, standard deviation, skewness, and kurtosis, and second order metrics including 24 gray level run length matrix (GLCM), 16 gray level run length matrix (GLRLM), 16 gray level size zone matrix (GLSZM), 5 neighboring gray-tone difference matrix (NGTDM), and 14 gray level dependence matrix (GLDM) [30,31,32,33,34]. We used all 95 textual features with 2 demographic features, age and sex, chosen for machine learning model development for the prediction of biomarkers and survival analysis. Details of the definitions and calculations of these features have previously been reported [35,36]. Details of 95 features were reported in Appendix A.

### 2.6. Statistical Analysis

To find the best parameter setting of the machine learning model, we applied an optimization search grid algorithm on a support vector machine (SVM) [37], multilayer perceptron (MLP) [38], XGBoost [39], and RandomForest [40] classifiers with 7-fold cross-validation using our internal dataset. The performance of the developed models was then assessed in the external testing dataset. Each classifier having the best performance on the validation set was selected. Only the results of the best classifier were reported hereafter. Receiver-operating characteristic (ROC) curves were generated and area under the curve (AUC) was estimated for models from FLAIR features.

Kaplan–Meier analysis [41] and Cox proportional hazards (CPH) [42] were performed to study the prognosis of 18-month survival. CPH were compared to the best machine learning classifier. The DeLong test [43] was used to calculate the 95% confidence intervals (CI) of AUC and two-sided *p*-values. A P-value smaller than 0.05 was considered statistically significant.

## 3. Results

### 3.1. Clinical Characteristics of Patient Population

A total of 251 patients were reviewed. Patients were excluded if they had insufficient MR image quality (motion artifact, *n* = 23), prior surgeries involving the tumoral bed (*n* = 16), or were treated with radiotherapy previously (*n* = 12).

Our final patient cohort consisted of a total of 208 patients. The mean ± standard deviation of age (years) was 56 ± 15 with a median age of 56. Among our cohort, 130 were male and 78 were female. The breakdown of the WHO glioma grades (2/3/4) were 28/56/124.

### 3.2. Testing Dataset

In our testing dataset, the cohort size was 31 patients with a mean ± standard deviation of age (years) of 54 ± 14 with a median age of 54. Among our cohort, 21 were male and 10 were female. The breakdown of the WHO glioma grades (2/3/4) were 3/0/28. IDH-1 wt comprised 27/31 patients and MGMT methylation was in 22/31 patients.

### 3.3. Auto-Segmentation and Prediction of Biomarkers

The DSC of the generated mask compared to the ground truth mask on FLAIR was 0.93 on the external testing dataset (Figure 1) (Table 1). We conducted a 7-fold cross-validation on the training dataset to analyze the variability in the AUC. The prediction for IDH-1 status had a performance of AUC 0.88 (95% CI: 0.84–0.93) in the training dataset and 0.93 (95% CI: 0.90–0.97) in the testing dataset using RandomForest (Table 2) (Figure 2). The prediction for MGMT status achieved an AUC 0.59 (95% CI: 0.51–0.68) and 0.62 (95% CI: 0.54–0.71) on training and testing datasets, respectively, using RandomForest (Table 3) (Figure 2).

### 3.4. Auto-Segmentation and Prediction of Survival

Survival data were available in a subset of patients within the training dataset (*n* = 89). A total of 150 patients were lost to follow-up, which included transfer to an outside hospital, hospice care, or no documented death in medical records. Survival prediction of <18 months demonstrated AUC of 0.75 (95% CI: 0.65–0.85) in the training dataset using MLP, while the CPH model achieved AUC of 0.53 (95% CI: 0.40–0.65; *p* < 0.05) (Figure 3). Kaplan Meier curve of survival estimation is shown in Appendix A. The comparisons of SVM, MLP, XGBoost, and RandomForest classifiers on the validation set were reported in Figure 4.

Figure 1 shows an axial image from a preoperative MRI showing a 75-year-old man with WHO grade IV glioblastoma with a biomarker profile of *IDH1* wildtype and *MGMT* methylation. Using FLAIR imaging (A), a volume of interest (VOI) was generated using a voxel-based signal intensity threshold method subsuming the entire region of FLAIR hyperintensity (B) used as a ground truth mask. The predicted mask from U-Net deep learning algorithm (C) is shown for comparison.

## 4. Discussion

Automatic segmentation of gliomas has been extensively studied in the past, most notably in studies from the recent BraTS trials, where large databases were assessed with successful segmentation with DSCs ranging from 0.74 to 0.92 [44,45,46,47,48,49]. In our study, the DSC was 0.93, which matched higher-performing algorithms from the BraTS trial (2017–2020). However, our algorithm only used FLAIR sequences compared to the multiparametric approach used in the BraTS trials.

In our study, we explored four U-Net approaches: the ResNet50 [50] network and DenseNet121 network by using pretrained weights generated from ImageNet [51] and RadImageNet, respectively. Table 1 summarizes the performance of these four models on both internal validation and external testing sets. DenseNet 121 with the RadImageNet pretrained weights were selected as it performed best on the validation set. This was likely because RadImageNet pretrained weights have higher similarities to the glioma dataset than ImageNet weights, thereby achieving better performance on both validation and test set and demonstrating reproducibility by showing a narrower difference between validation and test sets.

Several biomarkers are widely accepted as important prognostic indicators in gliomas. In our study, we report radiomic prediction for IDH-1 status with AUC of 0.88. Prior multimodal radiomic studies have predicted IDH-1 status with AUCs ranging from 0.86–0.90 [10,11,13]. We showed suboptimal predictability for MGMT methylation status with an AUC of 0.62. Prior studies have shown the prediction of MGMT status with AUC as high as 0.85 in multiparametric models [7,8,9]. The discrepancy in the performance of our FLAIR-only model to prior multiparametric approaches suggests that adding features from additional sequences likely increases the predictive ability for MGMT methylation status. Methylation of the MGMT gene has been associated with longer overall survival and favorable prognostic indicator of response to temozolomide and radiotherapy [52]. However, subsequent studies have reported conflicting results on the prognostic implication of MGMT methylation independent of therapy [53].

In a subanalysis of our internal dataset, survival analysis demonstrated good predictability of survival of <18 months with an AUC of 0.75. Previously, Bae et al. demonstrated improved overall survival prediction (integrated AUC of 0.65) in patients with GBM when combining radiomic MRI phenotyping with clinical and genetic profiles [54]. A separate study by Choi et al. showed that MGMT methylation and IDH-1 status when combined with radiomic data, provided valuable prognostication of GBM with a mean overall survival of 25.5 months and integrated AUC of 0.73 [55]. Similarly, radiomics and gene expression data were used by Ma et al. to stratify patients with low-grade gliomas into low- vs. high-risk of progression to higher-grade tumors with an AUC of 0.79 [56].

Establishing an automated pipeline that can predict treatment response and prognosis in patients with glioma can guide treatment options and surgical/chemotherapy management, potentially serving as a noninvasive alternative to brain biopsy and tissue resection. External validation of this algorithm also introduces applicability of this algorithm across institutions, although a larger validation set and multiple institutions should be further assessed in the future.

Our study has several limitations. Firstly, our study is retrospective in nature, and consequently, our patient cohort had a skewed distribution consisting predominantly of GBMs, especially in our testing group. Although our study had a higher proportion of low-grade gliomas compared to the BraTS trial, a balanced cohort of WHO-grade tumors would increase the applicability of our algorithm for low-grade gliomas. Another challenge is the heterogeneous nature of gliomas, especially in GBMs, which vary across individual patients and spatially within each tumor. Thus, biomarker profiles may vary depending on the site of biopsy, even within the same tumor, and a comprehensive biomarker landscape may not be captured by biopsy alone. Further validation with larger external datasets and multiple institutions are needed to demonstrate the applicability of this algorithm. Another limitation is related to truncated survival analysis due to incomplete data related to the retrospective nature of this study. Survival data for many patients were not available due to undocumented death in medical records, loss of follow up and switching providers, among many others. We also did not have survival data for most of our external dataset; therefore, the model performance in survival prediction could not be independently tested.

In summary, we showed that our deep learning-based framework can detect and segment gliomas with excellent performance and can provide high prediction accuracy for IDH-1 and modest accuracy for MGMT. Early identification of these biomarkers provides several advantages to clinicians and patients by allowing prognosis prediction and informing treatment decisions. We hope to expand this study to include serial follow-up imaging to assess changes in glioma radiomics and heterogeneity correlation with biomarker status, and in particular in patients who received prior treatment, where a noninvasive assessment of biomarkers may be a promising diagnostic tool without the need for re-surgery.

## 5. Conclusions

Our deep learning-based framework can detect and segment gliomas with a DSC of 0.93 and provide an acceptable prediction of biomarkers and prognosis. If its potential is realized, our automated pipeline may be used as a noninvasive assessment of glioma characteristics with important prognostic and therapeutic implications.

## Figures and Tables

**Figure 1 cancers-14-04457-f001:**
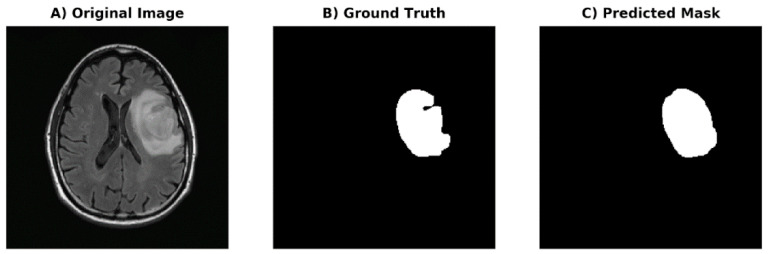
Manual segmentation and U-Net deep learning segmentation masks.

**Figure 2 cancers-14-04457-f002:**
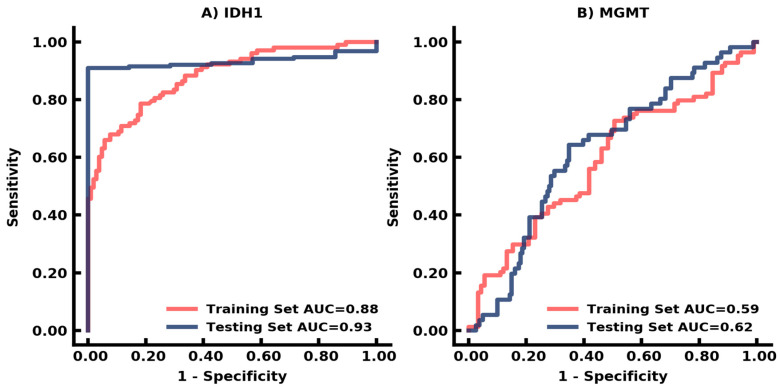
Biomarker prediction from U-Net deep learning algorithm. The receiver operating characteristic (ROC) curves demonstrate AUCs for (**A**) IDH-1, (**B**) MGMT.

**Figure 3 cancers-14-04457-f003:**
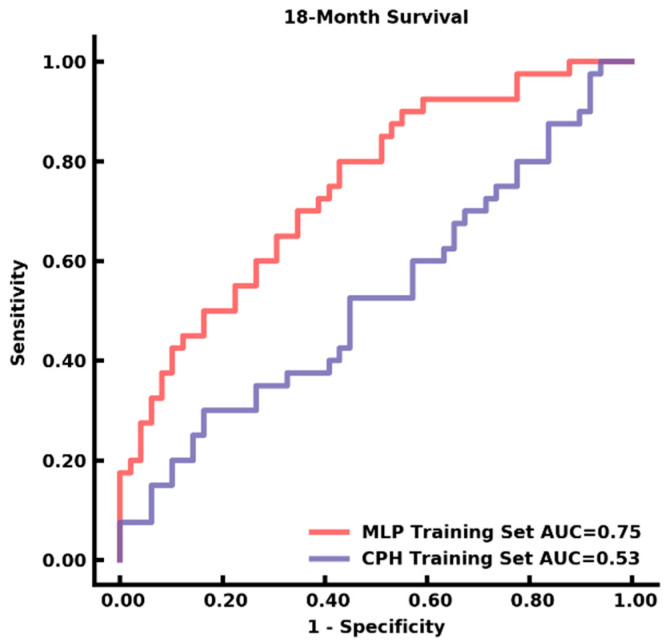
The 18-month survival prediction on the training set.

**Figure 4 cancers-14-04457-f004:**
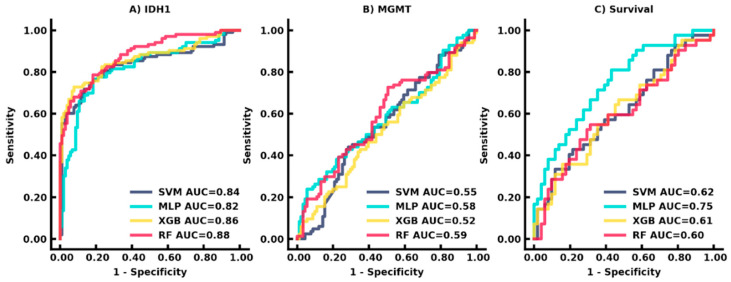
Comparison of four classifiers in the training dataset. Receiver operating characteristic (ROC) curves demonstrate AUCs for (**A**) IDH-1, (**B**) MGMT, and (**C**) 18-month survival prediction.

**Table 1 cancers-14-04457-t001:** U-Net performance.

Network	Pretrained Source	DSC on Validation Set	DSC on Test Set
ResNet50	ImageNet	0.94	0.83
ResNet50	RadImageNet	0.94	0.89
DenseNet121	ImageNet	0.92	0.83
DenseNet121	RadImageNet	0.96	0.93

**Table 2 cancers-14-04457-t002:** IDH-1 performance.

	Sensitivity	Specificity	Negative Predictive Value	Positive Predictive Value
Training Set	0.9	1	0.28	1
Testing Set	0.98	0.32	0.94	0.59

**Table 3 cancers-14-04457-t003:** MGMT performance.

	Sensitivity	Specificity	Negative Predictive Value	Positive Predictive Value
Training Set	0.63	0.65	0.83	0.38
Testing Set	0.45	0.68	0.57	0.57

## Data Availability

The data presented in this study are available on request from the corresponding author.

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
