# Peer review of "U-Net Based Segmentation and Characterization of Gliomas"

_cancers, 2022, doi:10.3390/cancers14184457_

Round 1
Reviewer 1 Report
This manuscript provides an interesting study into the use of MR images for biomarker and prognosis predictions for glioma patients. It is well written and presented clearly. I also appreciate that the study limitations are noted and that the authors plan to improve on these in future work.
3 very minor corrections before publication include:
1. It may be my downloaded version however the figures do not have A, B and C labels that correspond with the text. I would suggest checking all of the figures to make sure they are clearly labeled.
2. There are a few mistakes within the text. E.g. missing commas and changes in font sizes. A thorough read through for formatting, grammar etc is recommended.
3. The abstract states >18 months when it should be <18 months?
Author Response
Thank you for these helpful comments. Please see attached.

Reviewer 2 Report
Peer Review of: S. Kihira, X. Mei, et al. “Multicentric U-NET based Segmentation and Characterization of Gliomas”
For consideration in: MDPI Cancers
Review Date: August 28th, 2022
Review Summary
This reviewer eagerly read S. Kihira, X. Mei, et al. “Multicentric U-NET based Segmentation and Characterization of Gliomas” for consideration in MDPI Cancers. The reported study is an excellent example of how imaging in conjunction with deep learning could supplement traditional diagnostic workflows. The successful use of RadImageNet is promising, as are the results in general. All this said, the reviewer has six major concerns with the study and claims. Below, find a description of the major concerns, along with a variety of minor concerns that the reviewer believes could strengthen the manuscript. This reviewer looks forward to a revised version of the manuscript.
Major Concerns
1. What is meant by “external testing”?
This could be clearer in the manuscript. This reviewer surmised that the experiments were run in such a manner that the training and internal testing were performed on a cohort of patients from one institution/hospital/center, and that additional testing was performed in an “external” cohort of patients from another institution/hospital/center. If this is the correct interpretation, the study holds more weight than many of the single-institution manuscripts reported previously in the literature.
For example, describing what is meant by external testing in the manuscript where it is first mentioned (P1L22) would strengthen the appeal of the paper initially to audiences. The title of the manuscript might also be changed to highlight this. The review was confused whether the multicentric U-NET referred to a convolutional neural network that segments multiple diseases within a single patient, or whether it referred to multiple hospitals. This only became clear through reading the manuscript.
Finally, the authors should clearly name the multiple centers participating in the study in the methods.
2. Reproducibility and Feature Selection
Section 2.5 of the manuscript indicates that “a total of 95 texture features were extracted”, which were subsequently used to construct prognostic models as described in Section 2.6 after dropping features that were highly correlated. In the interest of reproducibility, the authors should list the final set of features for all three models generated, as well as the number of features in each model.
Related to reproducibility, the reviewer is concerned with the presentation of data in Figure 2. While the data is impressive. The reviewer suggests the authors include the training ROC curves, along with information about the variability given the training set (e.g., 95% confidence intervals for the curves). The authors could take the final set of features and train each model using 5-fold cross validation in the training set to obtain variability in the AUC. This would allow the reader to ascertain whether the change in performance in the “Internal” and “External” cohorts were due to random variation, or drift or systematic shift in the dataset (i.e., MRI acquisition differences, differences in populations, etc.).
3. Rates of Biomarker Status
As the reviewer was reading the manuscript and reviewing the data presented in Figure 2 and Tables 1-3, concerns arose around the claimed performance of the biomarker and prognostic models. For example, the NPV in the external dataset was 0.06, while the sensitivity was 0.44 and specificity was 1.0. Similarly, there appear to be either only 2 events for the survival classifier to identify, or all events collapse onto two threshold values.
These led the reviewer to wonder what the rates of mortality events, and rates of biomarkers within the populations studied. It is hard to assess how good the model actual is without this information. The authors should report the rates or numbers of each to allow the reader to weigh the claimed performance.
4. Survival Analysis
Related to #3 above, the reviewer believes a further survival analysis should be conducted. While identifying survival at 18 months may be beneficial, it is unclear whether the trends would be statistically significant. Do Kaplan Meier curves support the idea that the classifier is actually prognostic of 18 months survival?
The authors should also consider applying survival analysis (e.g., Cox Proportional Hazards) using the binary classification coming from the deep learning algorithm and compare the results with a traditional linear model using only patient demographic and disease markers available at diagnosis. By comparing these two, the authors could definitively demonstrate the benefit of their approach over standard of care.
5. Tradeoff between Imaging Modality and Biomarker Identification
P6L188-199: Can the authors speculate on the biological mechanism of the performance of FLAIR in predicting IDH-1 status but not MGMT status? What biological mechanisms lead to structural changes that lead to a prognostic model? On the other hand, if there is a tradeoff between MRI acquisition, what is the relative benefit of knowing IDH-1 versus MGMT status? Given that the latter can be used in selecting therapy, while the former is primarily for prognostic purposes at this time, it seems that acquiring images that can elucidate MGMT status is more important than IDH-1. The reviewer believes that addressing these questions would greatly improve the impact of the manuscript.
6. Data Access
The authors should strongly consider making the de-identified ground truth data and segmentation maps available via a public repository. The Cancer Imaging Archive would be a good venue for the data.
Minor Concerns
1. P2L48-49: The authors write “This is in particular helpful…in previously treated patients in whom histopathology data may not be readily available,” yet also write that “patients with prior chemotherapy or resection were excluded” (P2L76-77). The methodology of the paper does not support this claimed utility described in the introduction.
2. P2L79: Why was 18 months selected as a cutoff for poor vs good prognosis? This is never described in the manuscript. Is this a standard in the field of GBM? Or was it selected because of follow-up in the patient cohort? Or was it selected to optimize the discrimination of the survival classification? Clarity on this point would be helpful.
3. P2L88: Spell out acronyms TR/TE/TI.
4. P2L92: The input size was “256 x 256 x 3”. What are input as the three channels to the CNN? This could be clearer within the paper.
5. P3L96-98: The authors describe the use of DSC, which is an acceptable statistic for semantic segmentation. However, the description sounds like the authors are describing IoU. The authors could clarify this in the manuscript.
6. P3L103-104: The authors indicate that the ground truth was segmented by an expert neuroradiologist. Please list the level of expertise of the radiologist (years of practice).
7. P3L104-105: The authors indicate that “A total of 95 texture features were extracted from predictive masks generated by the aforementioned U-Net via the pyradiomics package in Python 3.8.10.” Were the features extracted from the predictive masks, or from the FLAIR images in voxels where the tumor is predicted?
8. P3L136-P4L145: Much of the information is duplicated in Tables 1-4. The authors might consider summarizing top-line takeaways.
9. P4L148-L153: This appears to be a description of Figure 1 without referencing Figure 1. Was this intended to be part of the figure caption? Similar comment for P5L157-159. Additionally, these blocks of text appear to indicate subfigures which are not labeled in the actual images.
10. P6L168-174: The segmentation performance is particularly positive, especially when considering the fact that only FLAIR imaging was used, and that data from multiple institutions/centers were examined. What do the authors attribute the high performance to? Is it RadImageNet? If so, they should stress this fact more fully. The authors compare the segmentation performance to the BraTS trial. The BraTS trial data includes FLAIR imaging; how does the authors’ segmentation network perform on this data?
11. P6L180: ImageNet should be cited.
12. P6L215-P7L224: The authors present a very good description of the limitations. The reviewer things the number of events might also be a limitation worth mentioning.
13. Discussion: The manuscript is an excellent demonstration of the utility of RadImageNet. While this is briefly mentioned in the manuscript, the reviewer thinks the authors should stress this point, particularly the improved performance and the quick turnaround of the study.
14. Use a consistent style for “UNET”. The reviewer saw at least three styles throughout the manuscript (e.g., UNET, U-NET, and U-Net).
15. Similarly, the authors should use a consistent style when referring to the Dice. The reviewer saw at least two styles throughout the manuscript (e.g., Dice, DSC).
Grammatical Items
In general, the authors should proofread the manuscript for grammatical issue. The most common issue encountered were missing articles prior to nouns. The reviewer noted the following non-exhaustive list of grammatical issues:
· P1L18: The article “The” is missing before “Deep learning-based U-Net”
· P1L42: There is an extra left parenthesis before T1c, e.g., “((T1c+), …”
· P2L65-66: “fully automated” enters the sentence twice. The reviewer suggests removing the second instance.
· P2L75: The reviewer suggests an Oxford style semicolon prior to “and 3) …”
· P2L78: Suggest replacing the “to” with an “and” to join the date ranges.
· P2L94: Suggest replacing “was” with “were” at the end of the sentence describing the CNN training parameters.
· P3L97: There is an extra space after the hyphen of “deep learning- based”.
· P3L129: Suggest adding “were” after “or”.
· P3L141: Suggest adding a missing comma after “AUC 0.56 and 0.62”
· P6L182-183: Suggest: “DenseNet 121 with the RadImageNet pretrained weights were selected…”
· P6L184: Suggest changing “and” to “than” at the end of the line.
· P6L218: Suggest adding “the” prior to “BraTS trial”.
· P7L251: Suggest adding a “.” At the end of the Informed Consent Statement.
· P7L257: Suggest replacing ‘”’ with a “.” At the end of the Conflict of Interest Statement.
Author Response
Thank you for these helpful and thoughtful comments. Please see attached.

Round 2
Reviewer 2 Report
The authors have addressed all this reviewer's concerns aside from how the grayscale image was converted to a three channel input. Are all three channels just set the same value? Consider clarifying.
Author Response
Thank you for your comment. We have attached our response letter and have edited our manuscript accordingly.
